# RNA Sequencing and Weighted Gene Co-Expression Network Analysis Highlight DNA Replication and Key Genes in Nucleolin-Depleted Hepatoblastoma Cells

**DOI:** 10.3390/genes15121514

**Published:** 2024-11-26

**Authors:** Hannes Steinkellner, Silvia Madritsch, Mara Kluge, Teresa Seipel, Victoria Sarne, Anna Huber, Markus Schosserer, Raimund Oberle, Winfried Neuhaus, Alexander V. Beribisky, Franco Laccone

**Affiliations:** 1Center for Pathobiochemistry and Genetics, Institute of Medical Genetics, Medical University of Vienna, 1090 Vienna, Austriaanna.b.huber@meduniwien.ac.at (A.H.); alexander.beribisky@meduniwien.ac.at (A.V.B.); franco.laccone@meduniwien.ac.at (F.L.); 2Vienna Doctoral School of Pharmaceutical, Nutritional and Sport Sciences (PhaNuSpo), University of Vienna, 1090 Vienna, Austria; 3Center for Pathobiochemistry and Genetics, Institute of Medical Chemistry and Pathobiochemistry, Medical University of Vienna, 1090 Vienna, Austria; 4Competence Unit Molecular Diagnostics, Center Health and Bioresources, AIT-Austrian Institute of Technology GmbH, 1210 Vienna, Austria; winfried.neuhaus@ait.ac.at; 5Department of Medicine, Faculty of Medicine and Dentistry, Danube Private University, 3500 Krems an der Donau, Austria

**Keywords:** nucleolin, RNA Seq, bioinformatics, DNA replication, ribosome biogenesis, cancer, human disease

## Abstract

Background/objectives: Nucleolin is a major component of the nucleolus and is involved in various aspects of ribosome biogenesis. However, it is also implicated in non-nucleolar functions such as cell cycle regulation and proliferation, linking it to various pathologic processes. The aim of this study was to use differential gene expression analysis and Weighted Gene Co-expression Network analysis (WGCNA) to identify nucleolin-related regulatory pathways and possible key genes as novel therapeutic targets for cancer, viral infections and other diseases. Methods: We used two different siRNAs to downregulate the expression of nucleolin in a human hepatoblastoma (HepG2) cell line. We carried out RNA-sequencing (RNA-Seq), performed enrichment analysis of the pathways of the differentially expressed genes (DEGs) and identified protein–protein interaction (PPI) networks. Results: Both siRNAs showed high knockdown efficiency in HepG2 cells, resulting in the disruption of the nucleolar architecture and the downregulation of rRNA gene expression, both downstream hallmarks of a loss of nucleolin function. RNA-Seq identified 44 robust DEGs in both siRNA cell models. The enrichment analysis of the pathways of the downregulated genes confirmed the essential role of nucleolin in DNA replication and cell cycle processes. In addition, we identified seven hub genes linked to *NCL*: *MCM6*, *MCM3*, *FEN1*, *MYBL2*, *MSH6*, *CDC6* and *RBM14*; all are known to be implicated in DNA replication, cell cycle progression and oncogenesis. Conclusions: Our findings demonstrate the functional consequences of nucleolin depletion in HepG2 and confirm the importance of nucleolin in DNA replication and cell cycle processes. These data will further enhance our understanding of the molecular and pathologic mechanisms of nucleolin and provide new therapeutic perspectives in disease.

## 1. Introduction

Nucleolin is a multifunctional protein localized mainly in the dense fibrillar and granular components of the nucleolus, accounting for 10% of the total amount of protein [1]. Intensive research has revealed a remarkable role of this essential protein in ribosome biogenesis, including ribosomal RNA (rRNA) processing, maturation and ribosome assembly. These processes are indispensable for cell growth and survival since the level of rRNA synthesis, mediated by RNA polymerase I (POL I), regulates the translation of mRNA into proteins [2,3]. In addition to its nucleolar localization, nucleolin is also found in the nucleoplasm, cytoplasm and cell membrane, where it is implicated in various functions depending on the cell type and the environmental conditions, including DNA replication and repair [4], chromatin remodeling [5], rRNA transcription and processing [6,7], mRNA turnover and translation [8,9,10,11,12,13,14,15] and viral entry and replication [16,17,18].

For this reason, nucleolin is a critical player in various crucial cellular processes, and its dysregulation may be involved in disease onset and the development of malignancies. To date, there have been very few reports of mutations or splicing variants of the *NCL* gene involved in diseases [19], although it is known that mutations in genes encoding proteins involved in ribosome biogenesis can lead to defects in rRNA transcription, which are often presented in ribosomopathies [20,21]. In addition, the overexpression of nucleolin and its increased localization on the cell membrane have been linked to increased proliferation and survival of cancer cells due to tissue invasion and angiogenesis. The oncogenic effect of nucleolin appears to be multifactorial, which reflects the multiple functions of this protein [22].

In addition to its role in cancer, nucleolin is also implicated in viral infections. It acts as a receptor for several viruses, including the respiratory syncytial virus (RSV) and rabbit hemorrhagic disease virus (RHDV), facilitating their entry into host cells [23,24,25,26]. This highlights the dual role of nucleolin in both promoting cellular growth and serving as a pathway for viral pathogenesis. Beyond cancer and viral infections, the dysregulation of nucleolin has been observed in other diseases, including autoimmune disorders [27,28]. Its involvement in immune response suggests that it may play a role in the pathophysiology of these conditions [29].

Nucleolin’s multiple functions, the role that it plays in tumor progression and viral infection and its surface localization make it a promising candidate for innovative treatment strategies, currently an area of active and intensive research. Its unique localization to the membrane surfaces of various cell types, including cancer and infected cells, opens potential avenues for targeted therapies. For instance, nucleolin-targeted therapies have been developed using aptamers and siRNA to selectively inhibit the function of nucleolin, leading to reduced tumor growth and improved treatment outcomes in preclinical models [30,31,32].

As human transcriptome data for nucleolin are sparse and due to the prevalent need to unravel the full spectrum of the molecular functions of this versatile protein, we used the human hepatocellular cell line HepG2 to investigate the impact of reduced levels of nucleolin. The aforementioned cell line was chosen due to previous studies indicating nucleolin’s involvement in hepatocellular carcinoma tumor progression [33,34]. We analyzed differentially expressed genes (DEGs) and performed Weighted Gene Correlation Network Analysis (WGCNA) on transcriptomic data of nucleolin knockdown cells using two different siRNAs to identify novel target genes and pathways at the genetic level. In addition, we identified hub genes, which may help identify therapeutic targets such as for cancer and viral infections.

## 2. Materials and Methods

### 2.1. Cell Culture and Transient Transfection

The hepatocellular carcinoma cell line HepG2 (#ATCC-HB-8065, LGC standards, Wesel, Germany) was cultured in Dulbecco’s modified Eagle’s medium (DMEM, #41966, Gibco, Life Technologies, Carlsbad, CA, USA) supplemented with 10% fetal bovine serum (FBS, #F9665, Sigma-Aldrich, St. Louis, MO, USA) and 1% penicillin/streptomycin (#15140122, Gibco). For the siRNA experiments, 6- and 24-well plates (Greiner Bio-One, Kremsmünster, Austria) were coated with rat tail collagen (#122-20, Sigma). Different siRNAs specific for human nucleolin (ON-TARGETplus SMARTpool and its four individual siRNAs NCL05, NCL06, NCL07, NCL08) and a non-targeting siRNA pool were obtained from Dharmacon. Subconfluent cells were plated 24 h in advance and then transfected with siRNA to a final concentration of 10 nM using lipid transfection reagent INTERFERin^®^ (#101000028, Polyplus, Illkirch, France) according to the manufacturer’s protocol, with slight modifications. After four hours, the transfection mixture was replaced with fresh medium, and the cells were cultured for another 48–72 h before being used for experiments. The viability of the cells was monitored with an EZ4U cell-viability assay (#BI-5000, Biomedica, Vienna, Austria).

### 2.2. RNA Extraction and qRT-PCR

Total RNA was extracted 48 h after siRNA transfection using the RNeasy Plus mini kit as per the manufacturer’s instructions (#74134, Qiagen, Hilden, Germany). An iScriptTM gDNA Clear cDNA Synthesis Kit (#1725035, Bio-Rad, Hercules, CA, USA) was used to synthesize cDNA from total RNA. For quantitative real-time PCR, the iTaq Universal Probes Supermix (#1725131, Bio-Rad, Hercules, CA, USA) was used. Quantitative real-time PCR analyses were carried out using the comparative Ct (ΔΔCt) method. GAPDH was used as the internal standard.

### 2.3. Western Blotting

For protein analysis, cells obtained from 6-well plates were suspended in RIPA lysis buffer (#R0278, EMD Millipore, Boston, Millipore, MA, USA) supplemented freshly with protease inhibitor cocktail (#P8340, Sigma Aldrich, St. Louis, MO, USA), incubated on ice for 20 min and lysed by gentle sonication using a Q700 sonicator with a chiller (Bioké, Leiden, The Netherlands). The lysates were then centrifuged for 10 min at 10,000× *g* and 4 °C. Protein concentration was quantified from the supernatant using the BCA Protein Assay Kit (Thermo Fisher Scientific, Waltham, MA, USA). Then, 25 μg of each protein sample was separated on 12% SDS–PAGE followed by transfer to a nitrocellulose membrane using the iBlot^®^ Dry Blotting system (#IB1001, Invitrogen, Carlsbad, CA, USA). Blocking was performed with 5% milk powder in 0.05% Tween 20 in PBS (PBS-T) for 1 h at room temperature. The primary antibodies used were anti-nucleolin (#GTX13541, Genetex, Alton Pkwy Irvine, CA, USA) and anti-GAPDH (#MAB374, EMD Millipore, Boston, Millipore, MA, USA), both at a 1:1000 dilution. Incubation with primary antibodies was carried out in 5% milk powder in PBS-T, with overnight rotating at 4 °C. The next day, incubation with mouse (#7076S, Cell Signaling, Danvers, MA, USA) and rabbit (#7074S, Cell Signaling) secondary antibodies was performed at 1:10,000 dilution as appropriate, also in 5% milk powder in PBS-T, for 1 h at room temperature. The blot was imaged using Clarity Western ECL Substrate (#1705061, Bio-Rad, Hercules, CA, USA) according to the manufacturer’s instructions and the ChemiDoc Touch Imaging System.

### 2.4. Immunofluorescence

HepG2 cells were grown on 8-well cell culture chamber slides (#94.6140.802, Sarstedt, Nümbrecht, Germany) overnight and transfected with NCL and NT-siRNA for 72 h. Then, the cells were treated with 50 ng/mL actinomycin D (#A9415, Sigma-Aldrich, St. Louis, MO, USA) or DMSO as a control. Two hours later, the cells were fixed with DPBS-buffered 3.6% formaldehyde and permeabilized with 1% Triton X-100 (#T8787, Sigma-Aldrich, St. Louis, MO, USA). After blocking for 45 min with 2% bovine serum albumin (BSA, #A9647, Sigma-Aldrich, St. Louis, MO, USA) in DPBS, primary antibodies directed against anti-nucleolin (#N2662-200, Sigma-Aldrich, St. Louis, MO, USA), anti-fibrillarin (#ab184817, Abcam, Cambridge, UK), anti-UBF-1 (#PA5-82245, Invitrogen, Carlsbad, CA, USA) and anti-B23 (#B0556-100, Sigma-Aldrich, St. Louis, MO, USA) were applied and incubated for 1 h at room temperature. After washing the cells three times with DPBS, the secondary antibodies (donkey anti-mouse IgG Alexa Fluor™ 594, #715-585-150, donkey anti-rabbit IgG Alexa Fluor™ 647, #711-605-152; Jackson ImmunoResearch, West Grove, PA, USA) were incubated for 1 h at room temperature. For immunofluorescence staining of fibrillarin, an additional incubation step was performed for 1 h with a recombinant anti-fibrillarin antibody conjugated to Alexa Fluor™ 488 (#ab184817, Abcam, Cambridge, UK).

For nuclear staining, the slides were embedded in ProLong™ Diamond Antifade Mountant including DAPI (#P36966, Invitrogen, Carlsbad, CA, USA). The slides were analyzed using a confocal microscope Leica SP8 (DMI6000, Leica microsystems, Vizsla, Germany) with LAS X 3.5 software and an HCX PL APO CS 40×/1.25 OIL PH3 UV (No. 506181; Leica Microsystems, Wetzlar, Germany) objective.

### 2.5. RNA Sequencing and Identification of Differentially Expressed Genes

Total RNA was extracted from three independent siRNA-mediated nucleolin knockdown experiments as described above. Sequencing libraries were prepared using the QIAseq^®^ Stranded mRNA Select Kit according to the manufacturer’s protocols (Qiagen). Libraries were QC-checked on a TapeStation (Agilent, Santa Clara, CA, USA) using a high-sensitivity DNA kit for correct insert size and quantified using a Qubit dsDNA HS Assay (Invitrogen, Carlsbad, CA, USA). Pooled libraries were sequenced on a NextSeq^TM^ 500 instrument (Illumina, San Diego, CA, USA) in 1 × 75 bp single-end sequencing mode. Approximately 22 million reads were generated per sample. Reads in fastq format were aligned to the human reference genome version GRCh38 (GRCh38 human reference genome downloaded from NCBI (https://www.ncbi.nlm.nih.gov/ accessed on 27 September 2018) with Gencode 29 annotations (human genome annotations downloaded from Gencode (https://www.gencodegenes.org/ accessed on 22 November 2018) using Spliced Transcripts Alignment to a Reference (STAR) aligner [35] version 2.6.1a in 2-pass mode. Reads per gene were counted by STAR, and differential gene expression was calculated using DESeq2 [36] version 1.34.0. Differentially expressed genes (DEGs) were defined as those with a fold change above 1.2 or below 0.83, with an adjusted *p*-value cut-off of 0.05. The principal component analysis plot and clustered heatmaps are based on variance-stabilizing transformed-count data generated with DESeq2 [36].

### 2.6. GO Enrichment and KEGG Pathway Analysis

Gene Ontology (GO) analysis is a common method for annotating genes and determining biological properties, such as molecular function (MF), and biological processes (BP). The most significant terms for MF and BP, ranked by *p*-value, are shown in bubble plots using SRplot (https://www.bioinformatics.com.cn/srplot accessed on 4 September 2024), an online platform for data analysis and visualization [37]. The Kyoto Encyclopedia of Genes and Genomes (KEGG) pathway enrichment analysis was used to investigate the role of DEGs in various pathways. The Gene Ontology tool Shiny GO 8.0 (http://bioinformatics.sdstate.edu/go/ accessed on 9 October 2024) was used for KEGG analysis. The most significant terms were ranked by adjusted *p*-values and presented in bar plots [38,39,40].

### 2.7. Co-Expression Network Construction and Identification of Hub Modules

Low expressed genes across all samples were filtered if the sum of the raw gene counts of all samples was less than 50. WGCNA [39] was then performed using DESeq2-computed variance-stabilizing transformation (VST) count data with the R package WGCNA version 1.70-3 following the tutorial at https://du-bii.github.io/module-6-Integrative-Bioinformatics/2019/Session5/Network_Inference_with_WGCNA.html (accessed on 22 November 2023).

The optimal soft-threshold power was 8, as this was the lowest integer power value where the scale-free topology fit index reached 0.85. Gene Tree clusters were pruned with a deepSplit (sensitivity) of 4. Hub genes are defined as the maximum adjacency (connectivity) per module, computed with the function chooseTopHubInEachModule. Modules with high correlation coefficients (>0.7 and <−0.7) were considered as relevant and selected for subsequent enrichment analyses.

### 2.8. Protein–Protein Interaction (PPI) Network Construction and Screening for Hub Genes

Search Tool for the Retrieval of Interacting Genes/Proteins (STRING) (http://string-db.org accessed on 14 October 2024) is a public online database of gene and protein interactions that helps users to easily access unique and wide-ranging experiments and predict interaction relationship information [40]. In this study, STRING was used to construct PPI networks of DEG products and the most significant WGCNA module genes. Therefore, interactions with a comprehensive score >0.4 were considered to be statistically significant. The PPI networks were then imported into CytoScape software 3.10.1 to screen for hub genes using the cytoHubba v0.1 plugin. The hub genes were screened using the degree algorithm and the top genes were ranked in descending order according to degree scores.

### 2.9. Statistical Analysis

The data were analyzed using GraphPad Prism software (version 10.3.1, San Diego, CA, USA), except for RNA-Seq analysis, for which the indicated software in the corresponding figure legends was used. Data are reported as the mean  ±  standard deviation.

## 3. Results

### 3.1. siRNA-Mediated Silencing of Nucleolin in HepG2 Cells Affects Nucleolar Morphology

In order to investigate the molecular and cellular functions of nucleolin, we established an siRNA-based nucleolin knockdown cell model using the hepatocarcinoma cell line HepG2. Four different targeting siRNAs directed against different domains of nucleolin and a pool of these siRNAs were tested for their ability to suppress the expression of nucleolin in HepG2 cells. The strongest decrease in nucleolin levels could be observed with the two specific siRNAs, NCL06 and NCL07, which were therefore selected for all further experiments. A non-targeting siRNA (NT) was used as a control, and transfection with all of the siRNA constructs had no effect on the viability of HepG2 cells, as confirmed by an EZ4U assay (Appendix A).

The two specific siRNAs resulting in the most efficient knockdown showed a decrease in nucleolin protein expression level after 72 h to 10.6% for NCL06 and to 4.5% for NCL07, respectively (Figure 1A). In this study, only samples with a protein knockdown level of at least 80% were considered for further experiments and data analysis. NCL06 and NCL07 were then used to investigate *NCL* mRNA expression and showed a similar knockdown efficiency after 48 h of siRNA transfection. *NCL* mRNA levels were reduced to 12.4% for NCL06 and to 10.8% for NCL07 (Figure 1B), showing the consistent knockdown of NCL expression at the protein and mRNA level.

To study the effect of the knockdown on the localization of nucleolin, NCL06 and NCL07, siRNA-transfected HepG2 cells were investigated by immunofluorescence staining. The visualization of nucleolin-depleted cells revealed a strong decrease in the signal compared to control cells (NT). In addition, the number of nucleoli was significantly reduced, and an altered morphology of the nucleoli could be observed. These nucleoli seemed to be bigger and have a ring-like structure (Figure 1C). Interestingly, this effect was more prominent in NCL06-transfected cells than in NCL07.

### 3.2. Nucleolin Knockdown Reduces POL I-Transcribed rRNA Levels and Disrupts Nucleolar Structure

Nucleolin has been linked to ribosomal biogenesis, was previously shown to interact with ribosomal RNA, and is therefore important for pre-ribosome assembly and maturation [6,41]. It has been reported that nucleolin knockdown in HeLa cells causes a strong decrease in *45S* rRNA [5,42]. To further investigate the impact of nucleolin on pre-ribosomal RNA processing, total RNA was extracted from nucleolin-depleted cells and the levels of mature RNAs (*5S*, *5.8S*, *18S*, *28S*) and *45S* pre-ribosomal RNA were assayed by qRT-PCR. In line with previous work, a significant decrease in both *28S* and *45S* rRNA was observed in nucleolin-depleted cells. In addition, the expression of *5.8S* and *18S* rRNA (Figure 2A) was also reduced. Notably, we found that *5S* rRNA levels of NCL knockdown cells remained unchanged. As *5S* rRNA is transcribed by RNA polymerase III (POL III), this finding confirms previous data showing that nucleolin only modulates POL I activity [43].

These findings warrant a deeper insight into the impact of nucleolin knockdown on nucleolar structure. To address this question, immunofluorescence experiments were carried out. Only cells transfected with NCL06 were used, as transfection with this siRNA showed more apparent effects in the nucleolar region (Figure 1C). The three main components of the nucleolus were stained using specific antibodies as follows: granular component (GC, anti-B23), dense fibrillar component (DFC, anti-fibrillarin), and fibrillar center (FC, anti-UBTF) (Figure 2B). In general, nucleolin is located in the DFC (fibrillarin) and GC (B23) of the nucleoli, as presented in Figure 1C. However, the knockdown of nucleolin leads to a remarkable modification of the nucleolar structure, as all three components showed reduced staining intensity as well as altered architecture. These changes in nucleolin-depleted cells are evident from the ring-like structure of the FC and GC as well as the less pronounced granular architecture of the DFC, with increased fibrillarin translocation into the nucleoplasm.

To further investigate the impact of strongly reduced nucleolin expression on POL I activity, the NCL06-transfected cells were treated with the POL I inhibitor actinomycin D (ActD) and stained with the same antibodies for the three main nucleolar components (Figure 2B). The inhibition of POL I in NT- and NCL06-transfected cells was accompanied by the formation of nucleolar caps containing UBF-1 and, more noticeably, fibrillarin. Interestingly, the POL I-inhibited cells showed a distinct translocation of B-23 from nucleoli into the nucleoplasm, which is further markedly increased in nucleolin-depleted cells. These data indicate that the reduced expression of nucleolin leads to the inhibition of POL I, which is associated with the inhibition of the transcription of rRNA.

### 3.3. Transcriptomic Analyses of Differential Gene Expression in Nucleolin-Depleted HepG2 Cells

To further investigate the general role of nucleolin in gene regulation and given the fact that transcriptomic data on nucleolin-depleted cells are sparse, RNA-Seq was employed. A total of 44 DEGs were identified in NCL06 and NCL07 compared to NT-siRNA-transfected cells, including 22 upregulated DEGs and 22 downregulated DEGs (Figure 3A).

The hierarchical clustering of the DEGs is shown in Figure 3B, and Appendix A lists the 22 downregulated and 22 upregulated genes. Volcano plots were used to assess the gene expression variation between the NT and NCL06/NCL07 knockdown groups, as shown in Figure 3C.

To validate the results from RNA-Seq, four genes (two upregulated and two downregulated) were selected from the obtained DEGs at random and the mRNA expression levels were analyzed by qRT-PCR (Figure 3D). The mRNA expression levels of *Heparan-α-Glucosaminide N-Acetyltransferase* (*HGSNAT*) and *Flap Structure-Specific Endonuclease 1* (*FEN1*) were downregulated, whereas *RNA-Binding Motif Protein 14* (*RBM14*) and *CEP295 N-terminal Like* (*CEP295NL*, *DDC8*) were upregulated in NCL06 and NCL07 siRNA-depleted cells. These observations are in line with the findings obtained by RNA-Seq analysis, confirming their accuracy and reproducibility.

### 3.4. Construction of Weighted Gene Co-Expression Modules

In addition, a WGCNA was performed, including all samples that were also subjected to DEG analysis (excluding low-expression genes). A total of 46 modules were created with WGCNA. The dendrogram clustered by the dissimilarity measure is shown in Appendix A, and the clustering of the eigengenes module is in Appendix A. Next, the association between gene modules and siRNA-mediated NCL-knockdown cells was analyzed. As shown in Appendix A, ten modules were significantly (*p*-value < 0.05) associated with NCL knockdown and four of them were identified as key modules with an absolute correlation r > 0.7. These modules included 72 genes, and hub genes were identified as RBM14 for the orange module (r = 0.95); phosphatidylinositol 4-kinase, type 2, and β (PI4K2B) for the dark turquoise module (r = 0.83); protein-l-isoaspartate O-methyltransferase domain-containing protein 1 (PCMTD1) for the pale turquoise module (r = 0.79) and chromobox 5 (CBX5) for the dark olive-green module (r = 0.73). RBM14 and PI4K2B, the hub genes from the most significant modules, were also found in the list of DEGs obtained from RNA-Seq analysis.

### 3.5. Functional Enrichment Analysis Reveals DNA Replication as Key Pathway in Nucleolin-Depleted HepG2 Cells

Functional enrichment analysis (KEGG and GO Term enrichment) of DEGs in the nucleolin knockdown cell model was conducted to investigate the potential biological role of the common list of DEGs. KEGG analysis revealed three major pathways with significant changes in gene expression. The most prominent pathway found to be significantly enriched was DNA replication (KEGG:03030), a key process in cell proliferation. In addition, the thyroid hormone signaling pathway (KEGG:04919) and cell cycle (KEGG:04110) were identified to also be significantly enriched, as presented in Figure 4A. Previous studies have shown that the separate analysis of up- and downregulated genes can be beneficial [44] and therefore separate KEGG enrichment analyses of up- and downregulated genes between the NCL06 and NCL07 versus NT knockdown groups were conducted. Interestingly, the upregulated DEGs were not enriched in any pathway, using the false discovery rate (FDR) < 0.05. Due to this finding, only downregulated genes listed in Figure 4B were used for further functional enrichment analyses.

KEGG analysis of downregulated genes revealed two major pathways with significant changes in gene expression. As expected, the most prominent pathways identified to be significantly enriched were DNA replication (KEGG:03030) and the cell cycle (KEGG:04110). In addition, central carbon metabolism in cancer (KEGG:05230), the biosynthesis of amino acids (KEGG:01230) and metabolic pathways (KEGG:01100) were identified as being significantly downregulated, as shown in Figure 4C. Further KEGG pathway analysis, using the more stringent adjusted *p*-value (AdjP), identified DNA replication and the cell cycle pathway to be significantly enriched, which highlights their sensitivity in the response to nucleolin depletion.

GO term enrichment analysis of downregulated genes was conducted to obtain deeper insights into the identified key pathways. Two relevant categories were used for the enrichment analysis: molecular function (MF) (Figure 4D) and biological process (BP) (Figure 4E). GO MF analysis revealed DNA replication origin binding (GO:0003688), magnesium ion binding (GO:0000287) and DNA-dependent ATPase activity (GO:0008094) among the most significantly downregulated processes. GO BP analysis of downregulated genes in NCL knockdown cells showed DNA replication initiation (GO:0006270) and DNA-dependent DNA replication (GO:0006261), both part of DNA replication, as well as cell cycle DNA replication (GO:0044786) and the child term nuclear DNA replication (GO:0033260), to be the most significant downregulated processes. The category net plot (cnetplot) used for the visualization of biological processes (Appendix A) and molecular functions (Appendix A) highlights the linkages of individual genes and GO terms.

In addition to the enrichment analysis of downregulated DEGs, we performed functional enrichment analysis with genes in key WGCNA modules (72 approved genes in orange, dark turquoise, pale turquoise and dark olive-green modules) that were highly correlated with NCL06 and NCL07 treatment (Appendix A). Notably, WGCNA enrichment analysis confirmed DNA replication and the cell cycle as key pathways in the nucleolin-depleted cells.

### 3.6. Construction of PPI Network and Module Analysis Identified Eight Hub Genes Related to Cancer

A PPI network was generated using the STRING database to investigate the association between the DEGs. All 44 DEGs (22 up- and 22 downregulated) were employed in order to consider all relevant gene associations. The PPI network generated by the online tool STRING 12.0 (https://string-db.org/, accessed on 14 October 2024), using a minimum interaction score of 0.4, resulted in a total of 40 nodes and 24 edges (Figure 5A). The network graph highlights the interplay between nineteen proteins, as listed in Appendix A, with the highest confidence in the PPI network for *minichromosome maintenance complex component 6* (*MCM6*), *minichromosome maintenance complex component 3* (*MCM3*), *cell division cycle* (*CDC6*), *flap endonuclease 1* (*FEN1*), *MutS homolog 6* (*MSH6*) as well as *paraspeckle component 1* (*PSPC1*) and *RBM14*. However, 21 of these nodes were singletons, meaning that there was no association with any other node.

The cytoHubba v0.1 plugin from Cytoscape software was used for the identification of hub genes within the PPI network. MCODE analysis was used to distinguish subnetwork clusters within the PPI network and identified the most significant hub genes (Figure 5B): *MCM6*, *MCM3*, *FEN1*, *MYB proto-oncogene like 2* (*MYBL2*), *MSH6*, *CDC6* and *RBM14*. Interestingly, WGCNA also revealed *RBM14* as the hub gene in the most significant module (orange).

## 4. Discussion

Nucleolin is one of the most abundant non-ribosomal nucleolar proteins and is also found in the nucleoplasm and cytoplasm, as well as on the cell surface [45]. Nucleolin is known to play a major role in multiple steps of ribosome biosynthesis [41], cell cycle regulation [42], growth [46], cell death [22] and signal transduction [47]. In addition, it is highly expressed in tumors and actively dividing cells [48,49]. Hence, several strategies have been developed to target nucleolin on the surface of cancer cells to block their proliferation, apoptosis and angiogenesis [50].

Despite nucleolin being extensively studied, the full extent of the role of this multifunctional protein is not yet fully understood. There is a pressing need to investigate nucleolin’s molecular function in cancer cells, making transcriptomic analysis with deep-sequencing a valuable approach. In this study, a nucleolin knockdown model was generated in HepG2 cells using two specific, different, separate *NCL* siRNAs. We performed deep-sequencing transcriptomic analysis, providing new insights into the molecular function of nucleolin. Both *NCL* siRNA-depleted cell lines (NCL06 and NCL07) displayed features consistent with the disruption of the hallmark functions of nucleolin. One such characteristic is the role of nucleolin in ribosome biogenesis, particularly in the transcription of rRNA genes. A decrease in the levels of pre-*45S* rRNA as well as *18S*, *5.8S* and *28S* mature rRNAs was observed in both nucleolin-depleted cell lines, confirming its involvement in rRNA synthesis, which is subsequently crucial for ribosome biogenesis. Furthermore, the depletion of nucleolin led to the disruption of the nucleolar architecture in line with findings in HeLa cells and human fibroblasts [42]. Taken together, these observations indicate that the nucleolin-deficient HepG2 cell lines are suitable to further investigate its molecular functions via RNA-Seq analysis.

Differential gene expression analysis and WGCNA were used to identify nucleolin-related regulatory pathways and possible key genes. Overall, 44 DEGs were found to overlap in both nucleolin-depleted cell lines compared to the NT control. While this small number of DEGs might be surprising given nucleolin’s broad range of functions, these findings are consistent with other transcriptomic analyses, where only about 0.1% of the genes present in the microarray were altered in nucleolin-depleted cells [42]. A qRT-PCR analysis of four randomly selected genes from the list of DEGs further confirmed the aforementioned observations. *NCL*, as expected, has shown the strongest downregulation among all DEGs.

KEGG and GO pathway enrichment analysis revealed DNA replication and the cell cycle as the most significantly altered pathways in nucleolin-depleted cells, which provides additional evidence to support the findings of previous studies [4,45,51]. DNA replication, occurring during the S phase of the cell cycle, is considered to control growth along with the cell cycle pathway. It has been reported that diminished levels of nucleolin in HeLa cells have a profound effect on the cell cycle, with cells rapidly accumulating in the G2/M phase [42]. However, another study using siRNA-mediated nucleolin knockdown in the chicken DT40 B cell line reported a partial G1-arrest [52], suggesting that the role of nucleolin may vary depending on the specific cell types as well as genetic backgrounds. Interestingly, both pathways are associated with oncogenesis, further confirming previous findings linking nucleolin to cancer.

The STRING database was used to construct the PPI network, and a topologic analysis identified eight hub genes in nucleolin-depleted cells. In addition to *NCL*, the hub genes included *MCM6*, *MCM3*, *FEN1*, *MYBL2*, *MSH6*, *CDC6* and *RBM14*. These proteins, encoded by the aforementioned genes, are implicated in DNA replication [53,54,55,56] and cell cycle progression [57,58,59] and are also known oncomarkers, including that of hepatocellular carcinoma [60,61,62,63,64,65,66]. Taken together, these findings further reinforce the link of nucleolin to a plethora of cellular processes, the dysregulation of which is associated with the onset of oncogenesis [63,67,68,69,70,71]. For instance, a nucleolin hub gene panel may be of use for tumor detection or determination of infection status [72,73,74]. In addition, reduced nucleolin levels coupled with decreased RNA levels may serve as a readout for a multitude of ribosomal disorders [75,76].

There are several limitations in this study. One limitation is the lack of a characterization of other cell types and tissues. Because the effects of nucleolin depletion were investigated exclusively in hepatocytes, the findings described in this study might vary in other cell types. Possible candidates that can be used in a follow-up study include cell lines that recapitulate pancreatic and colorectal malignancies as well as normal cell lines. In addition, only two hub genes (*RBM14*, *FEN1*) were validated in this study. The validation of the remaining hub genes is also a limiting factor and has to be addressed in future studies. Finally, the functional analysis of these hub genes would enhance the understanding of their role and interplay in the context of nucleolin function.

In summary, we demonstrated that an siRNA-mediated downregulation of nucleolin in HepG2 cells results in a substantial downregulation of genes implicated in DNA replication and cell cycle progression. In addition, a PPI network of hub gene products all implicated in cell proliferation was identified. These findings shed new light on the importance of nucleolin in the cell cycle, further strengthening its link to tumorigenesis, which can potentially open new avenues for therapeutic studies targeting this crucial protein.

## Figures and Tables

**Figure 1 genes-15-01514-f001:**
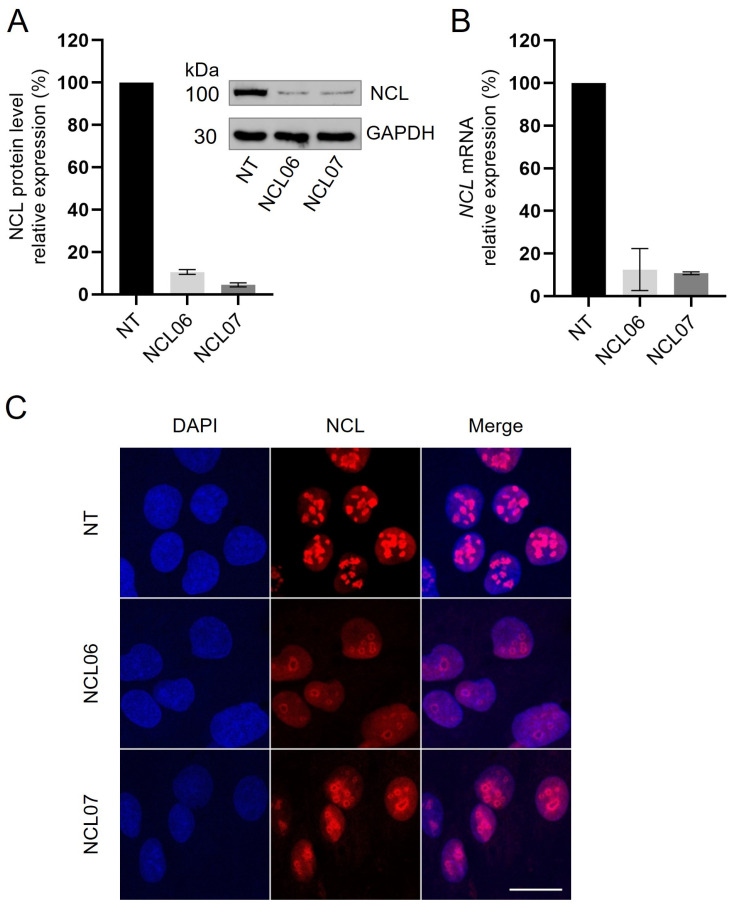
siRNA-mediated downregulation of nucleolin in HepG2 cells. HepG2 cells were seeded onto 24-well plates and were treated with 10 nM concentrations of two different nucleolin-targeting siRNAs (NCL06, NCL07) and 10 nM non-targeting-siRNA (NT). (**A**) Western blot analysis was performed on protein extracts from transfected cells after 72 h using anti-NCL antibodies for nucleolin detection. Data were normalized to GAPDH in reference to non-targeting control. Results are expressed as mean ± SD of three different experiments. (**B**) After 48 h of transfection, total RNA was extracted from cells and used for qRT-PCR with *NCL-* or *GAPDH*-specific primers. Data were normalized to *GAPDH* and are shown as % of NT-transfected cells. (**C**) HepG2 cells were seeded on glass slides and analyzed by immunofluorescence after 72 h of siRNA transfection (NT, NCL06 and NCL07) using anti-NCL antibody (red). DNA was counterstained with DAPI (blue). Scale: 20 µm.

**Figure 2 genes-15-01514-f002:**
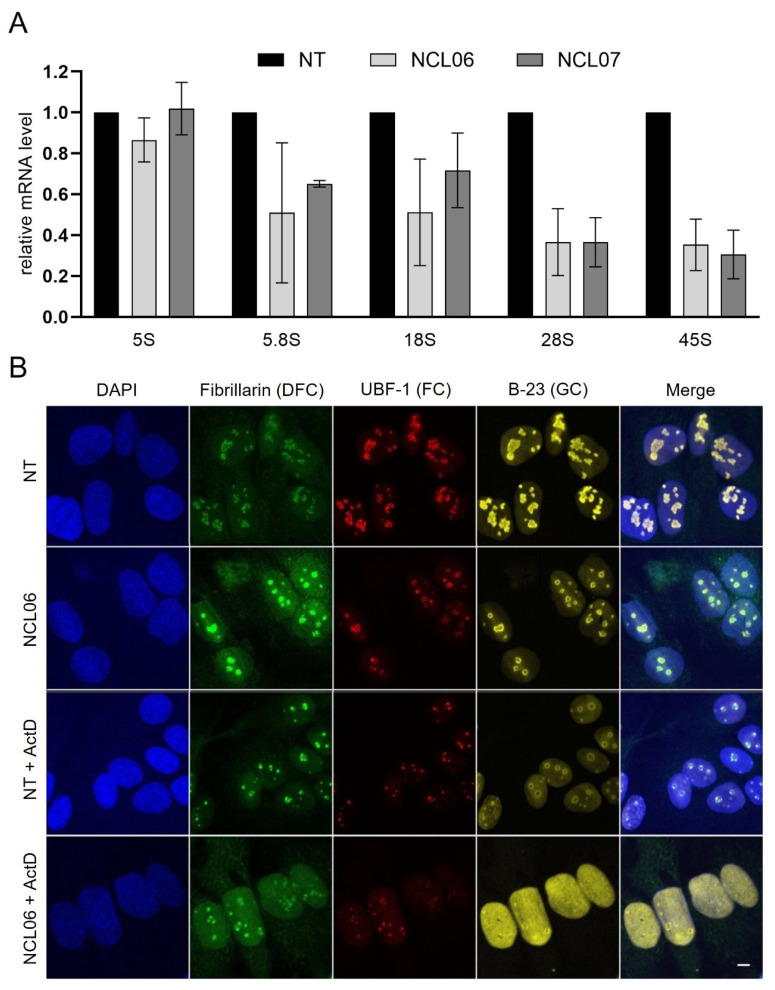
Nucleolar structure of nucleolin-depleted cells. (**A**) Total RNA from transfected cells was used to analyze expression levels of *5S*, *5.8S*, *18S*, *28S* and *45S* rRNA via qRT-PCR using specific primers. Data were normalized to GAPDH in reference to non-targeting control. Results are expressed as mean ± SD of three different experiments. (**B**) Representative images of NCL06-mediated NCL knockdown compared to NT-siRNA-treated cells stained with anti-B23 (GC, yellow), anti-fibrillarin (DFC, green) and anti-UBTF (FC, red). Nucleus was counterstained with DAPI (blur). Scale: 5 µm. For inhibition of POL I, cells were treated with 50 ng/mL ActD for 2 h and imaged with confocal microscope Leica SP8 (DMI6000, Leica microsystems).

**Figure 3 genes-15-01514-f003:**
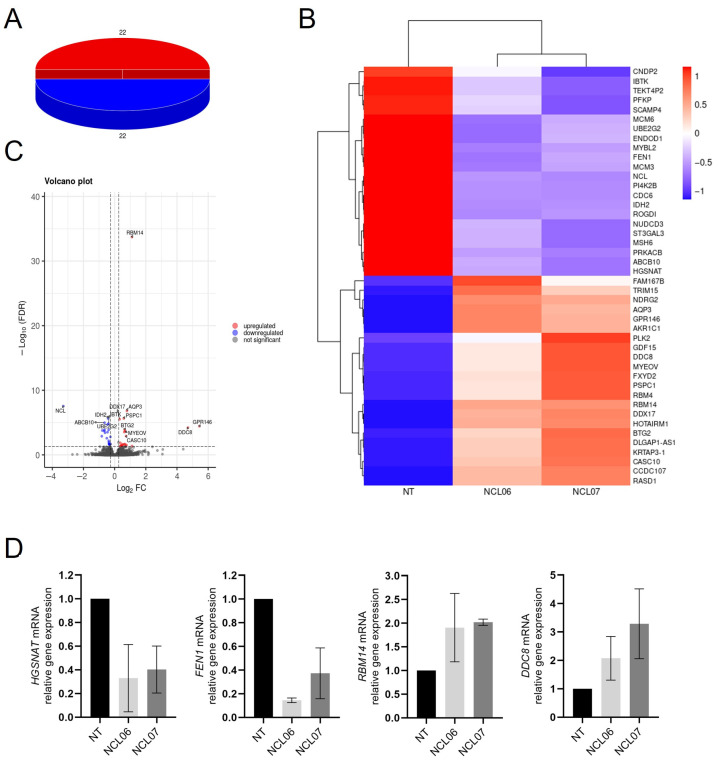
Differential gene expression in nucleolin-deficient cells. DEGs were screened with false discovery rate < 0.05 of DESeq2 using fold change above 1.2 or below 0.83. Total of 44 genes showed significant changes, of which 22 were upregulated and 22 genes were downregulated as presented by pie chart (**A**). Heatmap (**B**) and Volcano plot (**C**) of DEGs between NCL-06, NCL-07 and non-targeting siRNA. Red dots represent expression of genes in NCL-knockdown cells, which are upregulated compared to non-targeting siRNA (NT). Blue dots represent expression of genes in NCL-deficient HepG2 cells, significantly downregulated compared to non-targeting (NT) control. (**D**) qRT-PCR of four randomly selected genes to validate DEGs obtained from RNA Seq. Total RNA from transfected cells was used to analyze expression levels of *HGSNAT*, *FEN1*, *RBM14* and *DDC8* mRNA via qRT-PCR using specific primers. Data were normalized to *GAPDH* in reference to non-targeting control.

**Figure 4 genes-15-01514-f004:**
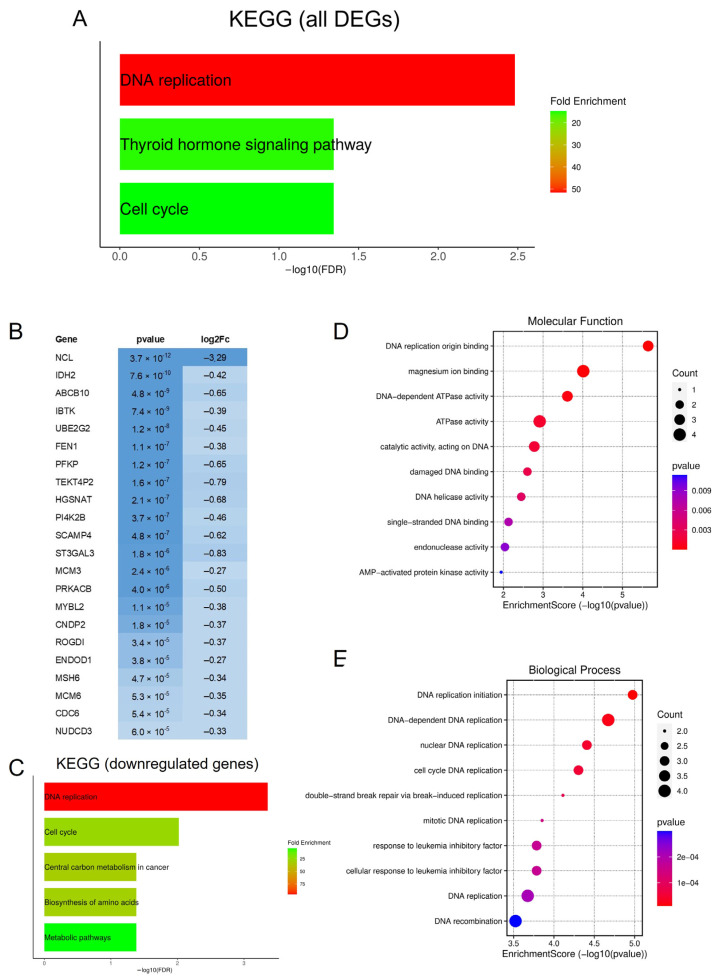
KEGG and GO enrichment analysis of downregulated DEGs in nucleolin-depleted cells. (**A**) KEGG pathway analysis of DEGs using false discovery rate (FDR) < 0.05. (**B**) List of 22 significantly downregulated genes from DESeq2 analysis (in blue). Intensity of blue color represents level of *p*-value and log2Fc of each gene in DEG list. (**C**) KEGG pathway analysis of 22 downregulated DEGs using FDR < 0.05. (**D**,**E**) GO enrichment analysis of downregulated DEGs in molecular function (**D**) and biological process (**E**) groups. Each term is ranked according to degree of significance, indicated by −log10 (*p*-value).

**Figure 5 genes-15-01514-f005:**
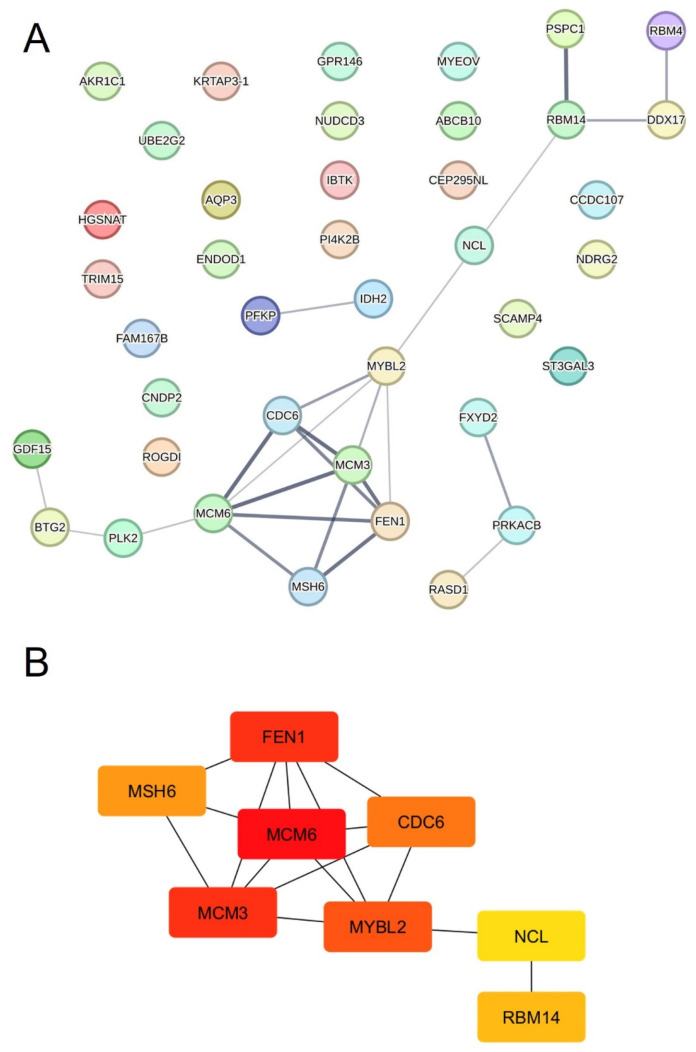
PPI network diagram and hub genes in NCL-depleted cells. (**A**) Protein–protein interaction analysis (STRING) from 44 DEGs; nodes represent proteins and edges represent PPIs. Thicker edge line corresponds to higher confidence score. (**B**) CytoScape with cytoHubba plugin and MCODE analysis was used to determine hub genes in NCL-depleted cells.

## Data Availability

The data presented in this study are available on request from the corresponding author.

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
