# Peer review of "RNA Sequencing and Weighted Gene Co-Expression Network Analysis Highlight DNA Replication and Key Genes in Nucleolin-Depleted Hepatoblastoma Cells"

_genes, 2024, doi:10.3390/genes15121514_

Round 1
Reviewer 1 Report
Comments and Suggestions for Authors This study provides valuable insights into nucleolin's role in DNA replication and cell cycle regulation, emphasizing its potential as a cancer therapeutic target. While the study's methods and focus are well-executed, broadening the analysis to other cell lines and validating the identified hub genes would enhance its impact.Expand cell models: Future studies should consider additional cell lines to improve understanding of the role of nucleolin in different tumors or tissue types.
In-depth functional analysis: Inclusion of functional assays for hub genes, such as RNAi knockdown experiments, would clarify their individual contributions to nucleolin-regulated pathways.
Expanding Pathway Analysis: Expanding functional enrichment analysis to include pathways beyond DNA replication and the cell cycle could reveal additional roles for nucleolin and potentially uncover more therapeutic targets.
The iThenticate shows 30% match, please reduce it. Comments on the Quality of English Language
Please check for any mistakes and improve.
Reviewer 2 Report
Comments and Suggestions for Authors
Here are some positive feedback to acceptance the manuscript.
This study uses RNA sequencing and Weighted Gene Co-expression Network Analysis (WGCNA) to investigate nucleolin's role in hepatoblastoma cells, revealing DNA replication and gene regulatory processes.
Differential gene expression, pathway enrichment analysis, and protein-protein interaction network creation are used in the study. A diverse approach considerably boosts the findings' depth and validity.
The identification of nucleolin depletion-associated genes and pathways advances cancer biology and may enlighten treatment options, given the link to DNA replication and cell cycle events.
Well-designed figures simplify the study's findings and help readers understand them.
The writers structure their findings in a logical and clear manner, making the work easy to follow and showing great topic knowledge.
Reviewer 3 Report
Comments and Suggestions for Authors
The authors present an experimental study where two different siRNAs are used to downregulate the expression of nucleolin in a human hepatoblastoma (HepG2) cell line.
It is an adequately structured study. The introduction is adequate, with a correct justification of the reasons for the study.
The methodology is adequate, with a detailed description of the experimental study process, which allows its reproduction.
The figures reported are adequate and clarify the results of the study, while helping the graphic representation of the same.
In the Discussion, which is too brief in my opinion, I miss a paragraph commenting on the clinical applicability of this research. All basic research should have a certain translational perspective, so the authors should comment on this aspect. They should also clarify whether, in addition to sequencing in hepatocytes, it could be performed in other cell types.
The references provided are current, and are adapted to the editorial needs of this journal.
